# Factors Impacting Range Hood Use in California Houses and Low-Income Apartments

**DOI:** 10.3390/ijerph17238870

**Published:** 2020-11-28

**Authors:** Haoran Zhao, Wanyu R. Chan, William W. Delp, Hao Tang, Iain S. Walker, Brett C. Singer

**Affiliations:** 1Indoor Environment Group and Residential Building Systems Group, Lawrence Berkeley National Laboratory, Berkeley, CA 94720, USA; haoranzhao@lbl.gov (H.Z.); wrchan@lbl.gov (W.R.C.); wwdelp@lbl.gov (W.W.D.); iswalker@lbl.gov (I.S.W.); 2National Centre for International Research of Low-Carbon and Green Buildings, Ministry of Science and Technology, Chongqing University, Chongqing 400045, China; haotang1994@cqu.edu.cn

**Keywords:** indoor air quality, cooking pollutants, kitchen ventilation, occupant survey, particulate matter, nitrogen dioxide, exposure mitigation

## Abstract

Venting range hoods can control indoor air pollutants emitted during residential cooktop and oven cooking. To quantify their potential benefits, it is important to know how frequently and under what conditions range hoods are operated during cooking. We analyzed data from 54 single family houses and 17 low-income apartments in California in which cooking activities, range hood use, and fine particulate matter (PM_2.5_) were monitored for one week per home. Range hoods were used for 36% of cooking events in houses and 28% in apartments. The frequency of hood use increased with cooking frequency across homes. In both houses and apartments, the likelihood of hood use during a cooking event increased with the duration of cooktop burner use, but not with the duration of oven use. Actual hood use rates were higher in the homes of participants who self-reported more frequent use in a pre-study survey, but actual use was far lower than self-reported frequency. Residents in single family houses used range hoods more often when cooking caused a discernible increase in PM_2.5_. In apartments, residents used their range hood more often only when high concentrations of PM_2.5_ were generated during cooking.

## 1. Introduction

Cooking is one of the largest sources of air pollutant emissions inside many homes. Gas cooking burners emit carbon monoxide (CO), nitrogen dioxide (NO_2_), formaldehyde (HCHO) and ultrafine particles and electric burners emit ultrafine particles in substantial quantities [1,2,3,4,5,6,7]. NO_2_ from gas cooking burners may commonly result in indoor concentrations that exceed the threshold of 100 ppb over 1 h that is used in the U.S. ambient air quality standard [8,9]. Belanger et al. [10] reported that exposure to higher levels of residential NO_2_ was associated with asthma severity and a meta-review by Lin et al. [11] found that gas cooking and higher NO_2_ exposure were each associated with increased risk of asthma and higher NO_2_ was associated with current wheeze. High temperature cooking activities (e.g., frying and broiling) contribute odors and pollutants including hazardous organic gases, polycyclic aromatic hydrocarbons, and fine and ultrafine particles [12,13,14,15,16,17,18,19,20]. In addition to the health effects associated with higher exposures to these pollutants (e.g., [21]), a study in Hong Kong [22] reported a dose–response relationship between lifetime exposure to cooking fumes and lung cancer. Both gas burners and cooking generate water vapor that may contribute to excess indoor moisture and associated problems if not adequately managed [23].

Venting range hoods and combination “over the range” (OTR) microwave/exhaust fans mounted above the cooktop are designed to remove some fraction of the emitted pollutants to outdoors before they mix into the air volume of the kitchen and throughout the home. Several modeling and experimental studies have examined the effectiveness of range hood use to reduce cooking-related indoor air pollution [5,8,9,24,25,26,27,28,29,30].

It is not clear how many US homes have venting range hoods because their presence has not been investigated in large surveys of home appliances [31,32]. Published data from indoor air quality (IAQ) surveys indicate that venting range hoods are installed in many homes in California. For instance, a study of 644 low-income Latina households in Salinas found that 65% of the homes with gas stoves were equipped with functional kitchen ventilation devices [33]. A survey-based study of 1448 detached houses built in 2003 reported that 80% had range hoods exhausting to outdoors and 4% had downdraft ventilators, while 13% had recirculating range hoods and 3% had no range hood or did not know [34]. A recent survey of occupants in 2781 homes built since 2003 and that have gas cooking burners found that 76% had a venting kitchen range hood or OTR, 12% had a recirculating range hood or OTR, and 4% reported not knowing what they had [35]. While we did not find data reported for large IAQ studies from other US states, the authors understand from discussions with building professionals throughout the US that the presence of kitchen ventilation varies both across and within states.

Venting range hoods are required in several building codes and standards. For example, ANSI/ASHRAE Standard 62.2 for residences requires a minimum on-demand airflow rate of 50 L/s or 100 cfm at a maximum sound rating of 3 sones [36]. The Energy Star program requires kitchen ventilation consistent with Standard 62.2 and additionally requires a minimum efficiency of 2.8 cfm/W and a maximum sound level of 2.0 sones for range hoods with power consumption less than 75 Watts [37]. Though venting range hoods are not required by the International Mechanical Code [38] or International Energy Conservation Code [39], they are required by many US states and cities. Starting with the 2007 update to the Title 24 Building Code—specifically in the Building Energy Efficiency Standards (BEES) that comprise Part 6 of the Code—California has required all newly constructed residences and major renovations to have kitchen ventilation equipment in concordance with the requirements of Standard 62.2 [40].

Venting range hoods help with IAQ management only if they are used when cooking occurs [5,41]. To quantify IAQ benefits from using range hoods, it is important to know how frequently and under what conditions they are operated during cooking. In many studies, range hood use has been estimated based on participant self-reporting. Studies have inquired of generic use (yes/no) and sometimes queried the frequency or reasons for using or not using the devices. For example, in the study of 1448 detached houses in California built in 2003, 28% of survey respondents reported using a kitchen exhaust fan when cooking with cooktop burners but only 15% reported exhaust fan use when cooking with an oven [34]. In the web-based survey of occupants in 2781 California homes built since 2003, 34% of households reported using their range hoods during cooking always or most of the time, 30% reported occasional (sometimes) use and 32% reported rarely or never using a hood [35]. In another California study, 34% of 372 homes reported using their range hoods during cooking, with higher frequencies during dinner and more use with longer cooking duration [42]. Higher resolution information on self-reported range hood use is available from daily activity logs recorded in some IAQ studies. In a study of 132 Canadian homes, Liu and Wallace found that only 13% of households reported range hood use during cooking events in winter and use decreased to 10% of cooking events in summer [43].

The goal of the study reported here was to assess actual range hood use based on monitoring of cooking activities and range hood operation in occupied homes. This study presents an analysis of data collected over weeklong periods in 54 houses and 17 apartments which were recently constructed or renovated in California. Data were analyzed to determine the frequency of range hood use during part or all of the cooking events with a focus on the effects of the following parameters: (1) cooking burner(s) used (cooktop, oven or both); (2) home type (house or apartment); (3) range hood type (conventional hood or OTR); (4) cooking duration (minutes of burner use); (5) self-reported usage; and (6) fine particulate matter (PM_2.5_) emissions during cooking. We also investigated whether the rate of range hood use in a home was associated with any household or equipment characteristics.

## 2. Materials and Methods

### 2.1. Overview of Data Collection

This study used data collected from two recent field studies of ventilation and indoor air quality in California homes with natural gas cooking appliances. All homes had natural gas cooktop burners and at least one oven, but some ovens were electric. All homes had venting range hoods and dwelling unit mechanical ventilation systems installed to satisfy state building code requirements. The Healthy, Efficient New Gas Homes (HENGH) study collected data in 2016–2018 in 70 single, detached houses that were built in 2011–2017 [35,44,45]. A second study collected data in 2018–2019 in 23 apartment units at 4 properties constructed or renovated in 2013–2017 [46,47]. The apartments were occupied by income-qualifying households and participants affirmed that they used their gas cooking burners on a daily or almost daily basis. HENGH homes had a mix of venting range hoods (*n* = 32) and OTRs (*n* = 38). All apartments had a venting range hood.

Each home was monitored for a one-week period and the residents were requested to keep windows closed and the dwelling unit mechanical ventilation system operating. Based on window monitoring and participant entries to daily activity logs, significant natural ventilation was used in only a small fraction of the houses and at least 5 of the 23 apartments. A number of indoor air pollutants were measured, including time-resolved fine particulate matter (PM_2.5_). PM_2.5_ concentrations were measured at 1-min logging intervals indoors and outdoors using MetOne photometers (Model BT-645 and ES-642) for houses, and 2-min logging intervals using DustTrak photometers (TSI Model 8530) for apartments. Operation of the cooktop and oven were monitored at 1.5-min intervals with iButton sensors (Maxim DS1922T) that were arranged on cooktops to enable detection of temperature changes caused by each burner, and at the oven vent. Range hood usage was monitored at 1-min intervals using a logging anemometer (Digisense WD-20250-22) placed at the air inlet or using a motor on-off logger (Onset HOBO UX90-004) placed close to the motor. Residents were asked to record occupancy and activities throughout each day of monitoring using a daily log sheet. A participant from each home completed a survey that asked about household demographics, satisfaction with environmental conditions in the home, use of ventilation equipment, and other activities that can impact IAQ. The surveys asked how often range hoods were used during cooktop use (for houses) or any cooking events (for apartments). Both of the field studies were conducted using protocols approved by the Lawrence Berkeley National Laboratory Human Subjects Committee.

The two field studies contributing data to this research were approved by Lawrence Berkeley National Laboratory’s institutional review board following US government regulations for research involving human subjects; the house study was protocol 318H003 approved 5/12/2015 and the apartment study was protocol 280H013 approved 11/19/2018.

### 2.2. Cooking Burner Events

Temperature data recorded by iButtons were analyzed to identify individual burner use events with specified start and end times. The start of a cooking event was identified by a rapid rise in temperature (Figure 1). A distinct threshold rate of temperature rise was specified to identify the start of cooking events in each home. Most thresholds were in the ranges of 0.6 to 1 °C/min for cooktop burners and 0.6 to 2 °C/min for ovens. The end of a cooking event was designated as the time when the burner temperature started to drop, with most decays being between 0.2 and 0.5 °C/min. Selection of the threshold value for each home was done by visual inspection.

Individual burner events that overlapped in time, or consecutive events that ended and started within 3 min of one another, were grouped into multi-burner cooking events. Each cooking event is defined by a start and stop time, burners used (CT for cooktop only, OV for oven only and CTOV for both), total minutes of CT burner use (e.g., 2 CT burners used for 10 min each is 20 burner-min) and total minutes of all burner operation. This includes the estimated full duration of OV use, not accounting for cycling of the OV burner. In some analyses, the term “any CT” is used to refer to cooking events involving cooktop burners (i.e., CT + CTOV).

### 2.3. Range Hood Operation

Data from the anemometers and motor on-off loggers were reviewed to identify and develop a data table of hood use events. There were a few cases when the loggers indicated high-frequency intermittent use at low speeds that we interpreted as continuous hood use. The timing of range hood use and cooking were compared to determine the amount of hood use (if any) during each cooking event. We defined “full use” as the range hood starting within 3 min of the start of the cooking burner and being used for >80% of the cooking time. Hood use was considered only if it overlapped in time with one or more cooking events; use that occurred independently of any cooking event was excluded from the analysis. In houses with OTRs, usage monitoring likely included incidences when the microwave was used to cook food and the fan was activated for that purpose, rather than for providing kitchen ventilation. The analysis only considered cooking events occurring during periods with valid range hood monitoring. Two apartments (identified as units 902 and 906 in the database) with range hoods operating continuously at low speed were excluded from the analysis. Three overnight uses of the oven in one apartment (unit 903) which were assumed to be done for heating, were also excluded from the analysis.

### 2.4. PM_2.5_ Event Identification

PM_2.5_ emissions were identified by applying a machine learning approach called Random Forest (RF) to the time-resolved PM_2.5_ concentration measured in the living room of each home, as described in detail elsewhere [48]. Briefly, the RF model was originally developed using a training dataset where the indoor and outdoor PM_2.5_ concentrations were collected from 18 California low-income apartments [49]. The model uses data features calculated from the indoor and outdoor PM_2.5_ concentrations to identify indoor emission events. A large number of classification decision trees were generated to express the full possible sequences of features to characterize a data point. The predominant classification of all the decision trees becomes the final prediction of the RF model. In this study, the RF model was applied to 2-min running average PM_2.5_ data for both houses and apartments.

The PM_2.5_ emission events identified by the RF analysis were reviewed visually to correct any obvious errors in start and end times.

A PM_2.5_ emission event was linked to cooking if it started during a cooking event and the PM_2.5_ emission duration was no more than 5 min longer than the cooking duration. This window is applied to account for the uncertainty in PM emission event end times identified by the RF model, and also for the time lag between the start of an emission event and an increase in PM concentration measured by the photometer. If a cooking event overlapped with more than one emission event, all of the emission events were considered to be associated with the cooking event.

### 2.5. Example Event Data

Example data from an event are shown in Figure 1. Temperatures were measured adjacent to four cooktop burners and the oven under the cooktop. Range hood operation was monitored by an anemometer and indoor PM_2.5_ was measured in the living room. Two cooktop burners were used in this event. The right front burner was started at 15:32 and stopped on or before 15:47 and the left front burner was started at 15:34 and stopped on or before 15:46. Much smaller temperature rises recorded by sensors at the oven and other two cooktop burners are assumed to result from the two front burners. Range hood use was determined to start at 15:35 and end at 15:57 based on the measured anemometer flow rate. Because the delay between range hood start and cooking start was less than 3 min and the range hood was used through the rest of the cooking event, it was considered as a cooking event with full range hood use.

### 2.6. Statistical Analysis

We investigated the influence of cooking parameters, ventilation equipment, and household characteristics on the fraction of cooking events in each home that had coincident range hood use and/or the fraction of total events across all homes in each group—houses and apartments—that had range hood use during some or most of the duration of each cooking event. The investigated cooking parameters were cooktop or oven use, total minutes of burner use and whether there was an identifiable, substantial increase in PM_2.5_ coincident with burner use; the latter was assumed to indicate a particle-generating cooking event. The studied ventilation equipment characteristics were conventional range hood or over the range microwave (OTR) and measured airflow and rated sound for highest and lowest settings. Home and household characteristics included floor area, number of occupants, occupant density, air exchange rate (total, per square meter, and per occupant), presence of senior or child someone with health condition that is impacted by air pollution, formal education, income, satisfaction with indoor air quality, satisfaction with air movement indoors, vacuum frequency, window opening frequency, and self-reported reasons for not using range hoods (forget, not need, ineffective, noisy).

Associations between range hood use and potential explanatory parameters were assessed in three different ways. Pearson’s chi-square test or Fisher’s exact test was used for categorical binary variables (e.g., whether range hood used or not categorized by cooking type). Wilcoxon rank-sum tests were applied for continuous variables, such as burner minutes for cooking events, to assess if the distributions differed between groups of events differentiated by categorical variables, e.g., in which range hoods were used or not used. An analysis of variance (ANOVA) test was applied to check relationships between two continuous variables, e.g., range hood use rate in each home vs. floor area. Relationships are considered very likely when the *p*-value is <0.05 and likely when the *p*-value is between 0.05 and 0.1. For continuous variables such as cooking burner-min, Wilcoxon rank-sum tests were applied to assess if the distributions differed between groups of events differentiated by categorical variables (e.g., in which range hoods were used or not used). Statistical analyses were performed using Stata version 15 (StataCorp, LLC, College Station, TX, USA).

## 3. Results and Discussion

### 3.1. Frequency of Cooking

Analysis of the iButton data found 607 cooking events in 57 single-family houses and 311 cooking events in 23 apartments. The distributions of total minutes of cooktop use and oven use per home per week at each hour of the day are shown in Figure 2 separately for houses and apartments. The mean and 10th–90th values of total cooking duration were 32 and 8–73 min for single-family houses and 40 and 8–56 min for apartments. In single family houses, the most cooking occurred during the late afternoon and evening (presumably around dinner) with a second mode during the morning, between 09:00 and 11:00. In apartments, cooking was more spread throughout the day with the peak occurring between 18:00 and 20:00.

Subsequent analyses exclude cooking events that occurred when range hood use was not monitored, cooking events that occurred in apartments with a range hood operating continuously, and overnight oven use that was presumed to occur for heating. The remaining data included 784 cooking events, with 574 events in 54 houses and 210 events in 17 apartments.

### 3.2. Fracion of Cooking Events with Range Hood Use by Home and Influencing Factors

Figure 3 shows the number of cooking events with no range hood use, any use, and full use in each house or apartment. On average, occupants in the apartments cooked slightly more meals (median = 9.0, mean = 12.4) than occupants in the houses (median = 7.5, mean = 10.6) during the weeklong monitoring period. About one-third (32%) of hood uses were considered full use, the remaining two-thirds of hood uses started with delays longer than 3 min and/or did not span 80% of the total cooking duration (see Appendix A for more details). Of the 37 houses and 16 apartments with five or more cooking events, 49% of houses and 63% of apartments used the range hood for less than 30% of cooking events and only two of each type used the range hood during more than 70% of cooking events. The mean rate was indistinguishable in houses and apartments.

There were no statistically significant associations between the fraction of cooking events with range hood use by home (henceforth, “rate”) and any ventilation equipment or household characteristics with analysis limited to homes with five or more cooking events (Appendix A). Similar analyses examining relationships between equipment or home characteristics and the likelihood of a range hood being used across all the cooking events in all homes with data, found several significant associations, as indicated in Appendix A. The most prominent factors were education and income level, with significantly higher rates of range hood use for cooking events in homes with higher income and education level. It is very important to note, however, that there were large and significant differences in education and of course income between the households in houses and the income-qualifying apartments. There also were statistically significant associations of lower range hood use when occupants were dissatisfied with air movement indoors (*p* < 0.01), open windows often (<0.01) and report that their range hood was ineffective (*p* < 0.01). In houses, range hood usage was likely lower when occupants said they sometimes do not use the range hood because it is not needed (*p* < 0.01) or self-assessed that their range hood was ineffective (*p* = 0.02).

Table 1 shows that residents in houses used their range hoods more frequently than residents in apartments when cooking with a cooktop (*p* = 0.006 for CT only and *p* = 0.01 for any CT). This difference may be connected to differences in range hood use by education and/or income level as noted previously. For example, residents in 46 out of 54 houses had bachelor degrees or higher, while only 2 out of 17 apartments had bachelor degrees or higher. Importantly, it is unclear which of the factors is driving higher rates of use, which could even be related to another factor that has not been quantified, e.g., the potential for a homeowner to select a preferred design or model of range hood or more familiarity with the equipment from a longer period of occupancy.

In houses with OTRs, some exhaust fan operation may result from automatic operation when microwaving food—with or without cooktop or conventional oven use—and thus not represent an intended use for ventilation. To investigate this, we repeated the analysis for houses with a regular range hood (i.e., not an OTR), and found the same result: residents in houses operated their standard range hoods more frequently than in apartments. No significant difference was observed in hood use between houses and apartments (*p* = 0.53) when cooking with the oven only. However, this finding is uncertain because there were very few cases of hood use during oven cooking events. Overall, the frequency of hood use during cooking with any burner was significantly higher (*p* = 0.03) in houses (36%) than in apartments (28%).

Table 1 shows that in single family house residents likely (*p* = 0.09) used a range hood or OTR more frequently when cooking with cooktop only, compared to oven only. In apartments, there was no discernible difference of range hood use with cooktop vs. oven use. Because ovens and cooktop burners can be used together in a cooking event, we compared full range hood use and any range hood use for different combinations of burner used, as shown in Appendix A. The overall results confirmed that for single family houses, hood use was likely more frequent when residents used a cooktop burner either alone or together with an oven burner, compared to use of the oven alone or with a cooktop.

### 3.3. Effect of Range Hood and Oven Type

Unlike in the apartments, which all had a similar set up of a conventional range hood over a cooktop with gas oven underneath, configurations varied in single family houses. Among the 54 houses with valid cooking and range hood use data, some had a regular range hood (*n* = 22), while others had an OTR (*n* = 32). Some houses had ovens located underneath the range hood (*n* = 32), while others had separate ovens located off to a side (*n* = 22). Some houses had gas ovens, while others were electric (7 of 32 of the underneath ovens, and 21 of 22 the separate ovens were electric). Statistical tests were performed to see if these differences are associated with the frequency of range hood use. Table 2 shows no discernible effect of range hood type in single family houses for cooking events that involved cooktop only. Table 3 shows it was likely (*p* = 0.09) that more frequent hood use was associated with ovens located underneath the range hood (36%), compared to separate ovens located off to a side (17%). However, this finding is uncertain because there were few cases of hood use when cooking with ovens. Additional analysis of hood use by oven location is provided in Appendix A.

### 3.4. Effect of Cooking Frequency

Moderate correlation was found between the rate of any range hood use and the number of total cooking events or cooktop uses in each house (Spearman coefficient of 0.36, *p* < 0.01 for total cooking events). However, no correlation was found between the rate of any range hood use and the number of any cooking burner events or of cooktop events in apartments (Spearman coefficient of 0.09, *p* = 0.73 for total cooking events).

### 3.5. Effect of Cooking Duration

Table 4 shows that range hood use was more frequent when cooktop burners were used during longer events. In houses, range hoods or OTRs were operated during 52% of events when cooktop burners were used for more than 20 burner-minutes, compared to 33% for 11–20 burner-minutes and 20% for 1–10 burner-minutes. Apartment residents also used range hoods slightly more frequently when the cooktop was used more than for 20 burner-minutes, but the association between hood use and cooktop use duration was not statistically significantly (*p* = 0.45). We also applied the two-sample Wilcoxon rank-sum test to the house data by sorting the cooktop use durations into two groups: “Hood used” and “Hood not used”. The mean (±s.d.) cooktop use duration in houses was 35 (±34) minutes for the “Hood used” group, and the mean was 20 (±20) minutes for the “Hood not used” group. Cooktop use duration was significantly different between these two groups (*p*-value <0.01). Applying the same analysis to data from apartments provides cooktop use durations of 29 (±41) minutes for the “Hood used” group and 23 (±22) minutes for the “Hood not used” group, with *p* = 0.38 indicating that the two distributions are not likely different. For cooking events using ovens, no apparent relationship between hood use and oven use duration was found (Appendix A).

### 3.6. Relationship of Actual Range Hood Use to Self-Reported Use

We compared actual range hood use and survey responses asking participants to self-report their range hood use habits in houses (Table 5) and apartments (Table 6). Note that the survey question and response options were somewhat different for the two studies. In the house study, participants were asked how frequently their range hood is used when cooking with a cooktop based on numerically-linked categories. In the apartment study, participants were asked about hood use during any cooking and given ordinal/categorical options. For consistency, Table 5 and Table 6 only consider cooking with cooktop burners (i.e., “any CT”). In both houses and apartments, actual hood use was higher in homes of participants that self-reported more frequent use, but actual use was much lower than self-reported use. For those reporting the most frequent range hood use—four or five out of five times in houses, or usually/always in apartments—actual hood use was only 45% and 36%, respectively. The difference is statistically significant among the houses (*p* < 0.01), and likely so among the apartments (*p* = 0.10).

To further elucidate if occupants are reliable reporters of range hood use, Figure 4 presents actual range hood use by self-reported hood use in houses and apartments. Actual hood use frequency was higher among households that reported to use their range hood most frequently, but there was wide variability and actual hood use was much less frequent than reported. This result indicates large bias when actual range hood use is estimated based on self-reporting in a survey.

Table 7 shows range hood use by cooktop burner-minutes and self-reported habits in all homes. These data reinforce the associations found between hood use frequency with both cooktop use duration and self-reported habits. Participants who claimed to use their range hoods always or most of the time actually used them about 60% of the time when using their cooktop for longer than 20 burner-minutes. However, the same households used their range hoods only 38% of the time when using their cooktop for 11–20 burner-minutes, and 26% of the time for 10 burner-minutes or less. Significant differences of actual hood use were found between the groups of households with different self-reported habits regardless of cooktop duration. For all levels of self-reported use, actual range hood use was significantly higher as cooktop use duration increased; this represents a rational prioritization to use the control more frequently as the potential hazard increases.

### 3.7. PM_2.5_ Emissions and Range Hood Use

There were 403 PM_2.5_ emission events identified for houses and 281 for apartments. Emission events varied vastly by duration and intensity, but the central tendency and range of values were similar among the houses and the apartments. The median PM_2.5_ emission duration was 16 min for the houses, and 14 min for the apartments. The 5th and 95th percentiles of PM_2.5_ emission duration were 10 and 42 min for the houses, and 4 and 52 min for the apartments. The highest 5-min PM_2.5_ concentration during emissions had a median value of 36 μg/m^3^ for the houses, and 37 μg/m^3^ for the apartments. The 5th and 95th percentiles of the highest 5-min PM_2.5_ concentration were 9 and 310 μg/m^3^ for the houses, and 9 and 250 μg/m^3^ for the apartments. We note the possibility that use of a range hood with high capture efficiency for particles theoretically could result in no substantial increase in PM_2.5_ in the space and thus no identified event. Limited data on range hood effectiveness for particles generated during cooking suggest that high capture could result when cooking at low to medium heat on the back burner, but not when cooking at high heat on a front burner [27].

Roughly 25% of cooking events in houses and 20% in apartments were linked with PM_2.5_ emissions (Table 8). In houses, a range hood was used for 58% of the cooking events with PM_2.5_ emission, and this was substantially and statistically significantly (*p* < 0.01) higher than range hood use when there was no PM_2.5_ emission detected (30%). In apartments, slightly higher use of range hoods when PM_2.5_ accompanied cooking (34% compared to 26%) was not statistically significant (*p* = 0.40).

Cooking events with associated PM_2.5_ emissions were categorized into two groups based on the peak 5-min PM_2.5_ concentration during emissions, and the association of this metric with range hood use is reported in Table 9. Higher peak PM_2.5_ concentrations could result from events with higher mass emissions, similar emissions being emitted into smaller spaces (noting that apartments are systematically smaller than houses), and/or slower mixing within larger homes. In houses, range hood use did not vary with the peak 5-min PM_2.5_ concentration. In apartments, however, range hood use was more frequent (56%) when the peak 5-min PM_2.5_ concentration exceeded 50 μg/m^3^, compared to only 28% when otherwise. But the differences were not statistically significant due to limited data. A possible reason for range hoods to be used less often in apartments when peak PM_2.5_ concentrations are lower is that emissions must be much smaller for peak concentrations to remain low in the apartments, which have much smaller volumes compared with houses; the smaller emission sources may not be as noticeable to residents.

Our analysis indicates that residents in single family houses used range hoods more often when cooking caused a discernible increase in PM_2.5_. In apartments, residents used their range hood more often only if high concentrations of PM_2.5_ (50 μg/m^3^) were generated during cooking. Even though we do not have additional information to explain this subtle difference in range hood use between houses and apartments, overall these results indicate that occupants in both types of homes took actions to address cooking emissions.

### 3.8. Limitations

A key limitation of our study is that the sample was not randomly drawn from the population, so findings may not apply more broadly. Our analysis was based solely on households living either in owner occupied houses that were built in recent years, or tenant occupied apartments that had been built or renovated in recent years. Relative to the general population of California, the households in the single, detached houses were skewed toward higher income and higher education as reported previously [44], while the apartments were recruited within low-income communities. The sample was not recruited to represent the diversity of cooking practices or even the diversity of cultures within California or the US. In addition, all of the households included someone that volunteered to participate for a one-week indoor air quality study, indicating at least a possibility for greater interest and attentiveness to IAQ hazards and controls than occurs in the general population.

Another important limitation is the small sample size, which limits the discernibility of some potential predictors that correlate with range hood use. In most cases, the relationship between range hood use and a factor was analyzed independently, rather than considering all the different factors together. We are limited by the small dataset to explore how all these factors in aggregation impact range hood use in homes.

The methods used to identify the start and stop times of cooking and range hood use were imprecise; this could have caused some errors in characterizing full or partial range hood use. Despite our best effort to visually inspect and correct the identification of cooking events, some ambiguity in the data remains. For example, it is difficult to estimate burner-minutes for cooking events that involved multiple burners. Future studies that can more precisely and certainly define cooking activities and link those to range hood use would advance understanding of how this residential IAQ control is used.

The linking of PM_2.5_ emission events to temporally proximate cooking events was uncertain because the source of PM_2.5_ may not be cooking related. The photometers used in the house and apartment studies were calibrated using a limited number of gravimetric filters that were collected. However, even after this calibration step, the adjusted PM_2.5_ measurements may have missed some cooking emissions, such as if the emitted particles were predominantly too small in diameter for the photometer to measure [20,50,51]. Future studies that confirm when PM is associated with cooking emissions would help in the understanding of how rationally people use their range hoods to control potentially hazardous contaminants.

## 4. Conclusions

We investigated range hood use for 784 cooking events in 71 homes including 54 single family houses and 17 low-income apartments constructed or renovated in recent years. Range hood use occurred more frequently with cooking in single family houses (36%) than in the apartments (28%). Range hood use by home generally increased with cooking frequency. In both houses and apartments, range hood use increased with cooktop use duration, but not with oven use duration. Participants who self-reported frequent use actually used their hoods more frequently; however, actual use was much lower than self-reported, with only 45% and 36% actual range hood use in houses and apartments where occupants self-reported use of always, usually, or most of the time. Residents in single family houses used range hoods more often when cooking events generated any level of PM_2.5_. In apartments, residents used the range hood more often only if high concentrations of particles were generated during cooking.

A better understanding of how range hoods are currently used in homes will help inform the potential benefits of adding sensing for automatic operation and improving awareness that range hoods should be used to reduce the population health burden from cooking emissions. The findings from this analysis are useful bases for future studies that aim to measure the impact of range hood use in reducing occupant exposure to indoor air pollutants in their homes.

## Figures and Tables

**Figure 1 ijerph-17-08870-f001:**
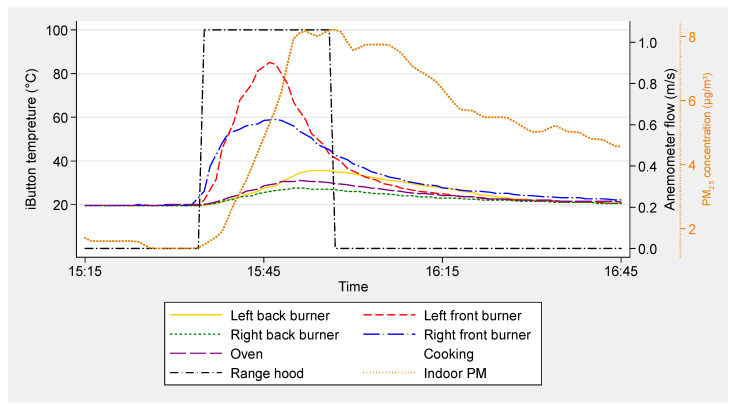
An example of full range hood use during a cooktop event with associated particulate matter emissions.

**Figure 2 ijerph-17-08870-f002:**
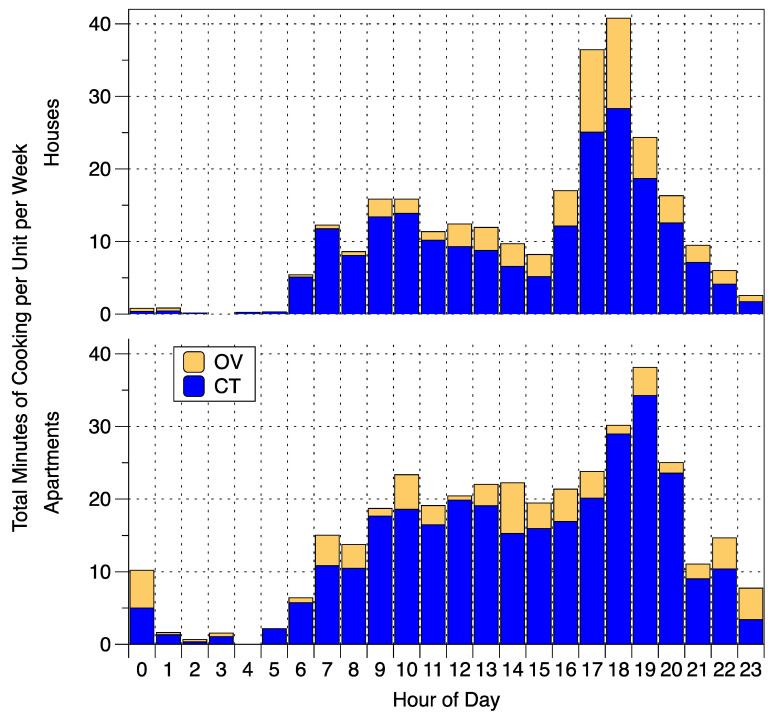
Distribution of total minutes of cooktop (CT) and oven (OV) use per home per week at each hour of the day in houses and apartments.

**Figure 3 ijerph-17-08870-f003:**
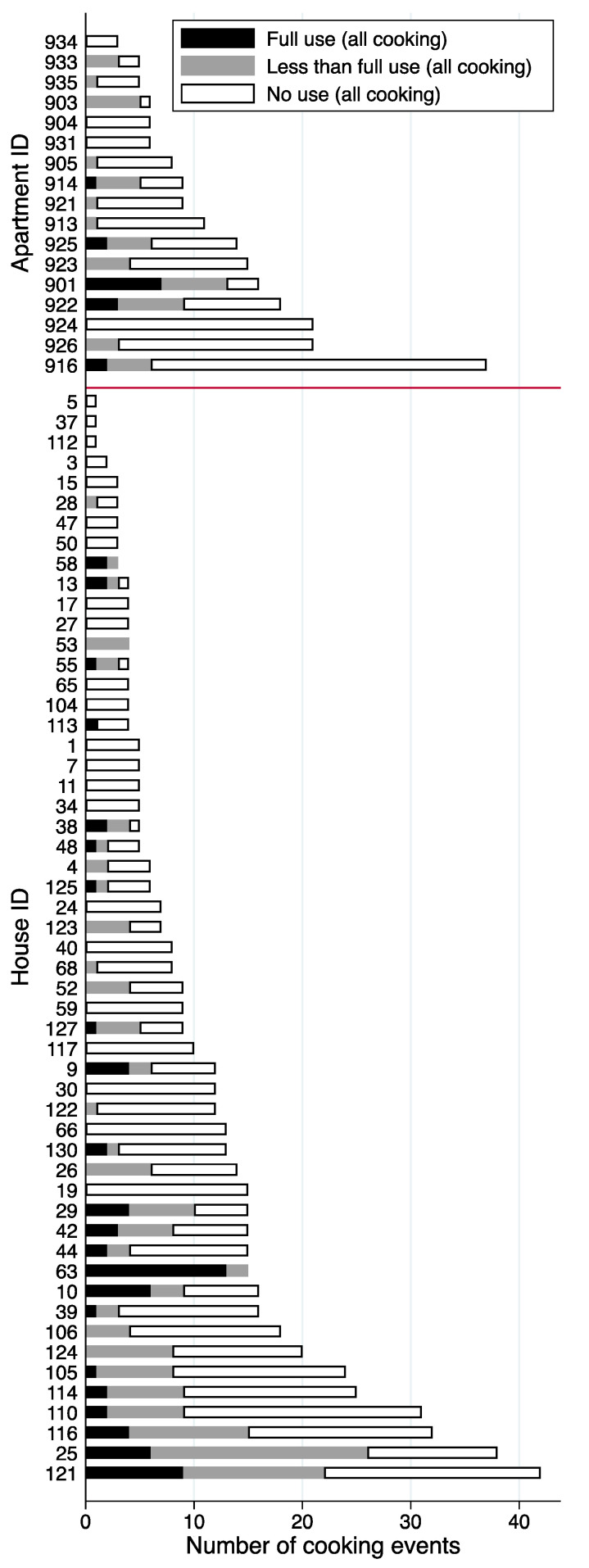
Range hood use for all cooking events by home.

**Figure 4 ijerph-17-08870-f004:**
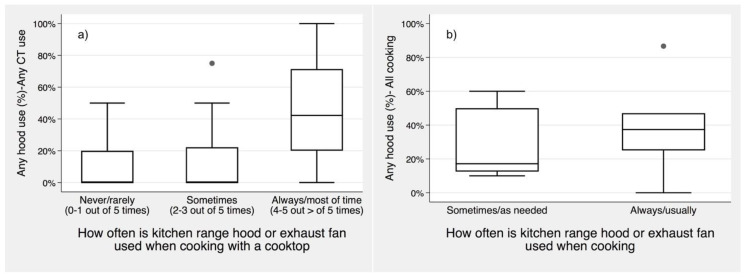
Distribution of range hood use in each home grouped by survey responses for (**a**) houses (any CT) and (**b**) apartments (all cooking types). Boxes show interquartile range (IQR), whiskers are limit values within 75th + 1.5IQR and 25th − 1.5IQR, and circles show all data outside of whiskers.

**Table 1 ijerph-17-08870-t001:** Range hood use by cooking type.

Cooking Type	Houses		Apartments		*p*-Value ^1^
	Cooking Events	Any Hood Use *n* (%)	Cooking Events	Any Hood Use *n* (%)	
CT only	487	182 (37%)	190	50 (26%)	0.006
OV only	48	12 (25%)	15	5 (33%)	0.53
CTOV	39	11 (25%)	5	3 (60%)	0.76
Total	574	205 (36%)	210	58 (28%)	0.03
*p*-value ^2^	0.09		0.56		

^1^ Chi-square test for hood use comparing two home types: houses and apartments. ^2^ Chi-square test for hood use comparing two cooking types: CT only and OV only.

**Table 2 ijerph-17-08870-t002:** Range hood use by range hood type in single family houses.

Range Hood Type	Cooking events—CT Only	Any Hood Use *n* (%)
Regular range hood	179	70 (39%)
OTR microwave	308	112 (36%)
*p*-value ^1^		0.54

^1^ Chi-square test for hood use comparing two hood types: regular range hood and “over the range” (OTR).

**Table 3 ijerph-17-08870-t003:** Range hood use by oven location in single family houses.

Oven Location	Cooking Events—OV Only	Any Hood Use *n* (%)
Under range hood	18	7 (36%)
Off to a side	30	5 (17%)
*p*-value ^1^		0.09

^1^ Chi-square test for hood use comparing the two types of oven location.

**Table 4 ijerph-17-08870-t004:** Range hood use by cooktop use duration.

Cooktop Use (Burner-Minutes)	Houses		Apartments	
	Cooking Events—CT Only	Any Hood Use *n* (%)	Cooking events—CT Only	Any Hood Use *n* (%)
1–10	143	29 (20%)	60	14 (23%)
11–20	139	46 (33%)	53	12 (23%)
>20	205	107 (52%)	77	24 (31%)
*p*-value ^1^	<0.01		0.45	

^1^ Chi-square test for hood use comparing different cooktop use durations.

**Table 5 ijerph-17-08870-t005:** Range hood use by self-reported use habit in houses.

Survey Response ^1^	Number of Houses	Cooking Events—Any CT	Any Hood Use *n* (%)	Cooking Events—All	Any Hood Use *n* (%)
Always/most of time (4–5 out of 5 times)	26	349	158 (45%)	371	166 (45%)
Sometimes (2–3 out of 5 times)	13	97	20 (21%)	109	22 (20%)
Rarely/never (0–1 out of 5 times)	13	70	11 (16%)	83	13 (16%)
I don’t know	0	0	0	0	0
No response	2	10	4 (40%)	11	4 (36%)
*p*-value ^2^		<0.01		<0.01	

^1^ Survey question: How often is a kitchen range hood or exhaust fan used when cooking with a cooktop? ^2^ Chi-square test with 1-side Fisher exact value for hood use comparing different survey responses.

**Table 6 ijerph-17-08870-t006:** Range hood use by self-reported use habit in apartments.

Survey Responses	Number of Apartments	Cooking Events—Any CT	Any Hood Use *n* (%)	Cooking Events—All	Any Hood Use *n* (%)
Usually or always	6	83	32 (39%)	92	33 (36%)
Sometimes/as needed	6	51	10 (20%)	57	14 (25%)
Rarely or never	0	0	0	0	0
I don’t know	3	46	6 (13%)	46	6 (13%)
No response	2	15	5 (33%)	15	5 (33%)
*p*-value ^2^		0.02		0.10	

**^1^** Survey question: How often is the range hood or exhaust fan used when cooking? **^2^** Chi-square test with 1-side Fisher exact value for hood use comparing the within first three survey responses.

**Table 7 ijerph-17-08870-t007:** Range hood use by self-reported use habits and cooktop use duration.

Cooktop Use (Burner-Minutes)	All Survey Responses			Never or Rarely (0–1 of 5 Times)			Sometimes or as Needed (2–3 of 5 Times)			Always/Usually or most of the Time (4–5 of 5 Times)			*p*-Value ^2^
	Cooktop Events—CT Only	Any Hood Use	% Use	Cooktop Events—CT Only	Any Hood Use	% Use	Cooktop Events—CT Only	Any Hood Use	% Use	Cooktop Events—CT Only	Any Hood Use	% Use	
1–10	194	41	21%	20	0	0%	41	6	15%	133	35	26%	0.01
11–20	175	55	31%	20	5	25%	42	7	17%	113	43	38%	0.03
>20	241	121	50%	24	6	25%	50	14	28%	167	101	60%	<0.01
*p*-value ^1^	<0.01			0.03			0.22			<0.01			

^1^*p*-value was calculated using Chi-square test for Any hood use among three burner-use duration groups within same response. ^2^*p*-value was calculated using Chi-square test for Any hood use among three groups with different responses within same burner mins.

**Table 8 ijerph-17-08870-t008:** Range hood use by cooking events with and without fine particulate matter (PM_2.5_) emissions.

PM_2.5_ Emissions	House		Apartment	
	Cooking Events	Any Hood Use (%)	Cooking Events	Any Hood Use *n* (%)
Yes	115	67 (58%)	41	14 (34%)
No	459	138 (30%)	169	44 (26%)
*p*-value ^1^		<0.01		0.33

^1^ Chi-square test for hood use frequency comparing cooking events with and without PM emissions.

**Table 9 ijerph-17-08870-t009:** Range hood use by cooking events with PM_2.5_ emissions.

Highest 5-min PM_2.5_ > 50 µg/m^3^	Houses		Apartments	
	Cooking Events	Any Hood Use *n* (%)	Cooking Events	Any Hood Use *n* (%)
Yes	58	30 (52%)	9	5 (56%)
No	57	37 (65%)	32	9 (28%)
*p*-value	0.19		0.23	

^1^ Chi-square test for hood use frequency comparing cooking events with peak PM below or above 50 μg/m^3^.

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
