# Peer review of "Factors Impacting Range Hood Use in California Houses and Low-Income Apartments"

_ijerph, 2020, doi:10.3390/ijerph17238870_

Round 1

Reviewer 1 Report

Comments:

  1. Line 118. Mention previous studies that have used the protocol.
  2. In the methodology, add images/diagrams of the hoods, houses, and apartments.
  3. Add dimensions of the houses/apartments and mention if there is any natural ventilation.
  4. Add more information about the study site: city, town, and coordinates.
  5. The title should mention that the impacts only refer to cooking events.
  6. Line 179. Add manufacture and model of the photometer.
  7. Line 165 to 174. The description looks like results. If not, then start the text mentioning that they are RF conditions.
  8. Line 183. Indicate that the RF estimates (not measures) indoor PM2.5.
  9. In the methodology, it is not clear which parameters were measured and which ones were estimated. Also, mention which are the RF inputs and outputs.
  10. Line 173. It is not clear how to interpret the results of PM2.5 concentrations, especially how to link these concentrations with the WHO and EPA guidelines.
  11. Why do the emissions obtained with RF have no units? Or it is not clear what they are.
  12. The difference between PM2.5 emissions and PM2.5 concentrations is not clear. Table 10 mentions emissions and PM2.5 concentration.
  13. Describe the equipment calibration in the methodology (photometer and anemometer).
  14. Mention the manufacturer and model of the gravimetric system used. Mention the filter type (manufacturer, diameter, and cyclone used). Also, mention the conditions and reference to obtain the filters PM2.5 mass
  15. The sample of 17 departments seems small, justify why that number of departments.
  16. Line 147. What is the value of high concentration?
  17. The study should include the monitoring of the cooking stacking. Please justify or improved this.
  18. There is no conclusion related to PM2.5 concentrations/emissions.

Author Response

We thank the reviewers for their thoughtful comments and suggestions. Below we provide a response to each specific comment from each reviewer. In response to comments by two of the reviewers that we should explore additional factors, consistent with the title, we made two substantial revisions to the text, adding analyses of additional equipment and household factors, as described in Section 2.6 and Section 3.2. We also moved the content of Table 1 to text in place of suggestion to remove Tables 2-3), updated references to the Zhao et al. paper and the database reporting results of an apartment IAQ study, on which this analysis relies, and did some section and subsection heading reorganization as requested.   

Reviewer 1 comments

  1. Line 118. Mention previous studies that have used the protocol.

Response: This sentence refers to the protocols submitted to the institutional review board that assures that research involving human subjects is conducted ethically and in alignment with U.S. regulations. A new protocol is created for each study and includes elements like consent forms and procedures for interacting with the study participants. The protocols for these studies have elements similar to those of other field studies conducted by LBNL and others, but neither has been previously used. 

  1. In the methodology, add images/diagrams of the hoods, houses, and apartments.

Response: It is a principle of human subjects regulations in the US to not publish information that could be used to identify participants unless they are informed up front that such information will be shared. And in general, it is discouraged to share - or even to collect - information that does not add value to the study questions. While we don’t doubt that some information could be gleaned by reviewing images of the home exteriors and even more so elements of the interiors, we did not feel that it would be of enough value to seek to obtain permission from participants to do so; so we cannot do so now. We note that the published datasets do include much information about the physical structures, households and equipment in the study homes, including e.g. floor area, number of stories for the houses, number of occupants, and both nominal and measured performance specifications for the range hoods. The published databases note whether the range hood in each home was a standard range hood or over the range microwave, the rated and measured airflows, and even brand and model number for most. Summary statistics are provided in the two recently published papers reporting on the field studies (Singer et al., 2019 and Zhao et al., 2020). 

  1. Add dimensions of the houses/apartments and mention if there is any natural ventilation.

Response: As noted in response to comment #2 the published databases (Chan et al., Dryad, 2019 and Zhao et al., Dryad, 2020) for the two field studies include specific data for each study home, including floor area, and the two published papers report summary information about many home characteristics. Regarding the question of natural ventilation, we have added a sentence to the Methods section describing the data to note the following: “Based on window monitoring and participant entries to daily activity logs, significant natural ventilation was used in only a small fraction of the houses and at least 5 of the 23 apartments.” 

  1. Add more information about the study site: city, town, and coordinates.

Response: As described in the response to Comment #2, we are not allowed under the approved human subjects protocol to provide individually identifiable information. When we published the dataset for the apartment study (Zhao et al., Dryad, 2020) we were cautioned to avoid providing information that would allow even an identification of the specific apartment buildings, as their identities are not relevant to the study data.  

  1. The title should mention that the impacts only refer to cooking events.

Response: We used the broader title because the submitted manuscript assessed the relationship of usage to housing type and range hood type, and also self-reported usage, in addition to cooking event parameters. In response to reviewer comments, we added analyses exploring associations to household characteristics including household income, size of home, occupant density, and whether any household members have conditions that could make them susceptible to air pollution. The analysis is described in Subsection 2.6 and results are presented in the new Subsection 3.2 in the Results and Discussion.  

  1. Line 179. [Add manufacturer and model of the photometer.]

Response: This information is provided earlier in the Methods, in Section 2.1: “PM2.5 concentrations were measured at 1-min logging intervals indoors and outdoors using MetOne photometers (Model BT-645 and ES-642) for houses, and 2-min logging intervals using DustTrak photometers (TSI Model 8530) for apartments.”

  1. Line 165 to 174. The description looks like results. If not, then start the text mentioning that they are RF conditions.

Response: The reporting of summary statistics about PM2.5 emission events associated with cooking has been moved to the Results, Section 3.7 (PM2.5 emissions and range hood use).  

  1. Line 183. Indicate that the RF estimates (not measures) indoor PM2.5.

Response: The mention of a measurement at this point in the text is accurate; it refers to the location of the photometer used to measure PM2.5. We realize that photometers make optical measurements of particles which are used to estimate mass-based PM2.5 using calibrations. But such details are beyond the scope of this paper; so we simply refer to the photometer data as PM2.5 measurements.

  1. In the methodology, it is not clear which parameters were measured and which ones were estimated. Also, mention which are the RF inputs and outputs.

Response: The methodology for identifying PM2.5 peaks using random forest approach is explained in detail in the cited paper (Tang et al., in review). We provide the working version of the manuscript to the reviewer as part of the resubmission. The text in the current paper is intended only as a brief overview of the approach. 

  1. Line 173. It is not clear how to interpret the results of PM2.5 concentrations, especially how to link these concentrations with the WHO and EPA guidelines.

Response: The text here refers to 5-min average concentrations. WHO and EPA guidelines and limits are for 24-h averages. So it is not appropriate to compare. We provide the 5-min averages as an indicator of short-term particle concentrations resulting from cooking. 

  1. Why do the emissions obtained with RF have no units? Or it is not clear what they are.

Response: It is unclear to us what results are being referenced in this comment. We note the number of PM emission events identified by the Random Forest method and note statistics of their durations (with units of minutes) and highest 5-min concentrations (with units of ug/m3).  

  1.  The difference between PM2.5 emissions and PM2.5 concentrations is not clear. Table 10 mentions emissions and PM2.5 concentration.

Response: The analysis distinguishes the impact of PM emissions from cooking events by the resulting concentration. The idea is that people may be more likely to use the range hood when cooking with PM emissions if those emissions result in higher concentrations of PM in the space. This could result from either a larger PM emission event or a smaller mixing volume. The caption correctly notes that the results apply to cooking events with PM emissions and divides by resulting concentration.   

  1. Describe the equipment calibration in the methodology (photometer and anemometer).

Response: These details are provided in the published papers about the field studies in which data were collected (Singer et al. 2019 and Zhao et al. 2020). There are too many details about primary data collection to repeat them all in the current paper.  

  1. Mention the manufacturer and model of the gravimetric system used. Mention the filter type (manufacturer, diameter, and cyclone used). Also, mention the conditions and reference to obtain the filters PM2.5 mass

Response: These details are provided in the published papers about the field studies in which data were collected (Singer et al. 2019 and Zhao et al. 2020). There are too many details about primary data collection to repeat them all in the current paper.  

  1. The sample of 17 apartments seems small, justify why that number of apartments.

Response: We agree that this is a small sample, limited to 23 apartments due to recruitment and scheduling challenges and a limited project budget and duration. And complete data were obtained for only 17 of the 23 apartments. For this reason, we note the following in the Limitations section: “Another important limitation is the small sample size, which limits the discernibility of some potential predictors that correlate with range hood use.” It is our hope that the methods reported here will be used by other researchers to conduct analyses on larger samples of homes. 

  1. Line 147. What is the value of high concentration?

Response: First, we presume this comment refers to line 174, as there is no mention of high concentrations in line 147. That said, we are still not sure we understand what is being asked by the reviewer. The analysis distinguishes the impact of PM emissions from cooking events by the resulting concentration. The idea is that people may be more likely to use the range hood when cooking with PM emissions if those emissions result in higher concentrations of PM in the space. This could result from either a larger PM emission event or a smaller mixing volume. The caption correctly notes that the results in Table 10 apply to cooking events with PM emissions and divides by resulting concentration.

  1. The study should include the monitoring of the cooking stacking. Please justify or improve this.

Response: The duration of the cooking event was from the first burner being turned on until the last burner was turned off. We also calculated the total burner minutes as the sum of the time that each burner was used. This is explained in Section 2.2: “Individual burner events that overlapped in time, or consecutive events that ended and started within 3 min of one another, were grouped into multi-burner cooking events. Each cooking event is defined by a start and stop time, burners used (CT for cooktop only, OV for oven only and CTOV for both), total minutes of CT burner use (e.g. 2 CT burners used for 10 min each is 20 burner-min) and total minutes of all burner operation.”

  1. There is no conclusion related to PM2.5 concentrations/emissions.

Response: The following text is included in the Conclusions: “Residents in single family houses used range hoods more often when cooking events generated any level of PM2.5. In apartments, residents used the range hood more often only if high concentrations of particles were generated during cooking.”

Reviewer 2 Report

I commend the authors are a great project and research article.  I have one minor recommendation. Even though the OV and CT are used throughout the article, Figure 2 is fairly early in the presentation and the figure does not mention what OV or CT are. You might edit the figure to present what OV and CT are again just for the convenience of the reader.

Author Response

Comment: I commend the authors are a great project and research article.  I have one minor recommendation. Even though the OV and CT are used throughout the article, Figure 2 is fairly early in the presentation and the figure does not mention what OV or CT are. You might edit the figure to present what OV and CT are again just for the convenience of the reader.

Response: We thank the reviewer for the positive assessment. We have modified the caption for the figure to explain that CT is for cooktop and OV is for oven. 

Reviewer 3 Report

In this papers the authors investigated range hood use for cooking events in homes including single families and houses, and low-income apartments, to assess actual range hood use based on monitoring of cooking activities and range hood operation. The parameters investigated were number of cooktop, oven or both, home type (house or apartment), range hood type (conventional hood or OTR), cooking duration, PM2.5 emissions.

Th paper is well written and data supports the conclusions about the hypothesis. Statistics were properly selected. Some small advices were addressed to the authors.

Figure 3: it is not really clear what are the numbers in the ordinate axis, please add a legend

figure 4: boxplots are pixelated please make use of a higher definition

Author Response

Reviewer 3 Comments

Comment: In this paper the authors investigated range hood use for cooking events in homes including single families and houses, and low-income apartments, to assess actual range hood use based on monitoring of cooking activities and range hood operation. The parameters investigated were number of cooktop, oven or both, home type (house or apartment), range hood type (conventional hood or OTR), cooking duration, PM2.5 emissions.

The paper is well written and data supports the conclusions about the hypothesis. Statistics were properly selected. Some small advices are addressed to the authors.

Comment: Figure 3: it is not really clear what are the numbers in the ordinate axis, please add a legend

Response: The numbers are the house and apartment identification codes. The axis title have been changed to “House identifier” and “Apartment identifier”.  

Comment: figure 4: boxplots are pixelated please make use of a higher definition

Response: Higher resolution image files for (a) and (b) portions of figure will be provided to the journal submission system. 

Reviewer 4 Report

The manuscript is dedicated to assessment of the actual range hood use based on monitoring of cooking activities and range hood operation in houses and apartments in California, USA. Attention is also paid to the study of the range hood use influence on the concentration of PM2.5 in the indoor air. The article is of scientific and practical interest. However, there are a number of remarks to the text of the manuscript.

The title does not quite match the results and goal of the manuscript (“The goal of the study reported here was to assess actual range hood use based on monitoring of cooking activities and range hood operation in occupied homes.”). The authors do not analyze the factors impacting range hood use. One could say that PM2.5 emission is a factor impacting range hood use, but this is not clearly explained in the manuscript.

Also, the aim of the study is not clear – why the investigation of the actual range hood use is important? What is the scientific purpose of the manuscript? It is necessary to formulate the research goal more clearly and to determine the relation between authors’ results and assessment of the risk to public health. Otherwise it is possible to relate described results to environmental research and public health or at least to SI topic “Health Risk Assessment Related to Environmental Exposure”.

Keywords: “nitrogen dioxide” – there is no analysis of emissions and concentrations of nitrogen dioxide in the manuscript.

In the introduction, if possible, it is necessary to pay more attention to the analysis of previous results on the health risks assessment of indoor air pollution formed by operation of gas cooking boilers, etc.

Please colorize all figures.

Line 44: it is necessary to decipher the abbreviation “IAQ surveys” (indoor air quality?)

Line 57: “ASHRAE Standard 62.2” => “ANSI/ASHRAE Standard 62.2”

Lines 165-174: these are the results, as there is lack of discussions on the causes of the PM2.5 concentrations variation. Particular attention should be paid to the analysis of the reasons of high PM2.5 concentrations.

Lines 214-216: it was already indicated in the “2. Materials and Methods” section.

It would be better to merge results and discussion sections. A broader discussion of the results obtained is required.

Lines 224-228: it is necessary to indicate which results are shown in Table 1. Interesting that hoods are used much more often in houses than in apartments (number of houses with 31-70% cooking events with any range hood use).

Tables 3 and 4 can be described in words in the text. There is no need to present these data in the form of a table.

Lines 274-278: what could be the reason for the lack of correlation between the rate of any range hood use and the number of any cooking burner events or of cooktop events in apartments? What could be the reason for the moderate correlation between the rate of any range hood use and the number of total cooking events or cooktop uses in each house?

Table 9: “Roughly 25% of cooking events in houses and 20% in apartments were linked with PM2.5 emissions”. What is the reason for such a low number of cooking events linked with PM2.5 emissions? Maybe those emissions are not detected because PM2.5 particles are effectively removed outside the building during the operation of the hood? Could the bad estimation by RF be the reason? This was not discussed in the article. It is also necessary to compare episodes with PM2.5 emissions when using venting range hoods and OTRs. The efficiencies of PM2.5 removing outside the buildings of these two systems are different, which can affect the amount of detected cooking events linked with PM2.5 emissions. Differences in cooking events linked with PM2.5 emissions in houses and apartments can also be attributed to the use of different venting and hood systems (all homes had venting range hoods and dwelling unit mechanical ventilation systems).

There is Subsection 4.1, but no Subsection 4.2 in the manuscript. If there is only one subsection in a section, then there is no need of such subdivision. “Limitations” can be moved into a separate section. The "Discussion" section can be merged with the "Results" section.

The data on the PM2.5 concentrations in the air require more detailed analysis and results description, because this part of the study is related to the effect on public health. In addition, the study does not show the average population of houses and apartments. What types of buildings are more populated and, consequently, which residents are more susceptible to the negative effects of PM2.5 air pollution? For a preliminary risk assessment, it is necessary to consider not only PM2.5 concentrations, but also the number of people exposed to high PM2.5 concentrations.

Author Response

Summary: The manuscript is dedicated to assessment of the actual range hood use based on monitoring of cooking activities and range hood operation in houses and apartments in California, USA. Attention is also paid to the study of the range hood use influence on the concentration of PM2.5 in the indoor air. The article is of scientific and practical interest. However, there are a number of remarks to the text of the manuscript.

Overview response: We thank the reviewers for their thoughtful comments and suggestions. Below we provide a response to each specific comment from each reviewer. In response to comments by two of the reviewers that we should explore additional factors, consistent with the title, we made two substantial revisions to the text, adding analyses of additional equipment and household factors, as described in Section 2.6 and Section 3.2. We also moved the content of Table 1 to text in place of suggestion to remove Tables 2-3), updated references to the Zhao et al. paper and the database reporting results of an apartment IAQ study, on which this analysis relies, and did some section and subsection heading reorganization as requested.   

Comment: The title does not quite match the results and goal of the manuscript (“The goal of the study reported here was to assess actual range hood use based on monitoring of cooking activities and range hood operation in occupied homes.”). The authors do not analyze the factors impacting range hood use. One could say that PM2.5 emission is a factor impacting range hood use, but this is not clearly explained in the manuscript.

Response: We used the broader title because the submitted manuscript assessed the relationship of usage to housing type and range hood type, and also self-reported usage, in addition to cooking event parameters. In response to reviewer comments, we added analyses exploring associations to household characteristics including household income, size of home, occupant density, and whether any household members have conditions that could make them susceptible to air pollution. The analysis is described in Subsection 2.6 and results are presented in the new Subsection 3.2 in the Results and Discussion. 

Comment: Also, the aim of the study is not clear – why the investigation of the actual range hood use is important? What is the scientific purpose of the manuscript? It is necessary to formulate the research goal more clearly and to determine the relation between authors’ results and assessment of the risk to public health. Otherwise it is possible to relate described results to environmental research and public health or at least to SI topic “Health Risk Assessment Related to Environmental Exposure”.

Response: It was an aim of the study to provide inputs for risk assessments of exposure to cooking burner pollutants and potential benefits of increasing range hood use. This was already noted in the Conclusions: 

A better understanding of how range hoods are currently used in homes will help inform the potential benefits of adding sensing for automatic operation and improving awareness that range hoods should be used to reduce the population health burden from cooking emissions. The findings from this analysis are useful bases for future studies that aim to measure the impact of range hood use in reducing occupant exposures to indoor air pollutants in their homes.”

We added the following sentence to the final paragraph of the Introduction, which presents the goal of the study:

The results of this study can be used in risk assessment for exposures to cooking-related pollutants and analyses of benefits of increasing range hood use.

Comment: Keywords: “nitrogen dioxide” – there is no analysis of emissions and concentrations of nitrogen dioxide in the manuscript.

Response: We include the term “nitrogen dioxide” because is an important pollutant emitted from gas cooking burners. 

Comment: In the introduction, if possible, it is necessary to pay more attention to the analysis of previous results on the health risks assessment of indoor air pollution formed by operation of gas cooking boilers, etc.

Response: We added two sentences and several references to the first paragraph of the Introduction to address this comment.

Comment: Please colorize all figures.

Response: We added color to Figure 1 but retained the distinct line types to enable someone who cannot see color to differentiate. We assess that the clarity of Figures 3 and 4 would not be improved by adding color as it is already easy to distinguish the data. 

Comment: Line 44: it is necessary to decipher the abbreviation “IAQ surveys” (indoor air quality?)

Response: We now define IAQ as indoor air quality at this point in the text. 

Comment: Line 57: “ASHRAE Standard 62.2” => “ANSI/ASHRAE Standard 62.2”

Response: this change has been made. 

Comment: Lines 165-174: these are the results, as there is lack of discussions on the causes of the PM2.5 concentrations variation. Particular attention should be paid to the analysis of the reasons of high PM2.5 concentrations.

Response: We moved this text to the Results section 3.7. We also added the following note: “Higher peak PM2.5 concentrations could result from events with higher mass emissions, similar emissions being emitted into smaller spaces (noting that apartments are systematically smaller than houses), and/or slower mixing within larger homes.” Detailed analysis of factors that impact PM concentrations resulting from cooking is beyond the scope of the present paper.  

Comment: Lines 214-216: it was already indicated in the “2. Materials and Methods” section.

Response: With appreciation for the ideal of not repeating anything, we feel that it is valuable here to clarify what data exclusion elements led to the final numbers of samples presented just following.

Comment: It would be better to merge results and discussion sections. A broader discussion of the results obtained is required.

Response: We made the change as suggested. The Limitations section is now included in Results and Discussion and Conclusions are now section 4.

Comment: Lines 224-228: it is necessary to indicate which results are shown in Table 1. Interesting that hoods are used much more often in houses than in apartments (number of houses with 31-70% cooking events with any range hood use).

Response: To address the subsequent comment, we eliminated Table 1 by bringing the main results into the text. 

Comment: Tables 3 and 4 can be described in words in the text. There is no need to present these data in the form of a table.

Response: When we tried to present these results in text form, we found that it didn’t save much space. And it also had the negative effect of making these important results less visible. So to address the suggestion of reducing tables to save space, we eliminated Table 1, and brought the key results from that table into the text. 

Comment: Lines 274-278: what could be the reason for the lack of correlation between the rate of any range hood use and the number of any cooking burner events or of cooktop events in apartments? What could be the reason for the moderate correlation between the rate of any range hood use and the number of total cooking events or cooktop uses in each house?

Response: Using the range hood more frequently in houses where cooking occurs more frequently is rational as the more frequent cooking could have a bigger impact on indoor air quality if not adequately controlled. The data collected don’t provide us any insights about the absence of this relationship in apartments.  

Comment: Table 9: “Roughly 25% of cooking events in houses and 20% in apartments were linked with PM2.5 emissions”. What is the reason for such a low number of cooking events linked with PM2.5 emissions? Maybe those emissions are not detected because PM2.5 particles are effectively removed outside the building during the operation of the hood? Could the bad estimation by RF be the reason? This was not discussed in the article. It is also necessary to compare episodes with PM2.5 emissions when using venting range hoods and OTRs. The efficiencies of PM2.5 removing outside the buildings of these two systems are different, which can affect the amount of detected cooking events linked with PM2.5 emissions. Differences in cooking events linked with PM2.5 emissions in houses and apartments can also be attributed to the use of different venting and hood systems (all homes had venting range hoods and dwelling unit mechanical ventilation systems).

Response: To answer the first question, there are many types of cooking that don’t produce substantial PM2.5. We thank the reviewer for the suggestion to note that effective range hood use could remove enough PM2.5 that it might not be seen in the space and thus not be recorded as an emission event. We now note this in Section 3.7.  

Comment: There is Subsection 4.1, but no Subsection 4.2 in the manuscript. If there is only one subsection in a section, then there is no need of such subdivision. “Limitations” can be moved into a separate section. The "Discussion" section can be merged with the "Results" section.

Response: With the creation of a combined Results and Discussion section, this subsection now appears in that section with a number of 3.8.

Comment: The data on the PM2.5 concentrations in the air require more detailed analysis and results description, because this part of the study is related to the effect on public health. In addition, the study does not show the average population of houses and apartments. What types of buildings are more populated and, consequently, which residents are more susceptible to the negative effects of PM2.5 air pollution? For a preliminary risk assessment, it is necessary to consider not only PM2.5 concentrations, but also the number of people exposed to high PM2.5 concentrations.

Response: We appreciate the reviewer’s points about the potential implications of different range hood use in apartments and houses, especially for cooking that release PM2.5. We conducted the study reported in this manuscript to support such analyses; but it would require too great of an expansion of the manuscript to incorporate such a risk assessment.

Round 2

Reviewer 1 Report

Comments and questions were answered.

The article has a defined objective, and the methodology section (especially the statistical method section) was improved. Section 3.2 helps to better understand the results obtained.

Reviewer 4 Report

The authors did a great job to improve the text of the manuscript. I believe that now all the obtained results are well-explained and the structure of the manuscript is reasonable.